# Characterisation of a *Staphylococcus aureus* Isolate Carrying Phage-Borne Enterotoxin E from a European Badger (*Meles meles*)

**DOI:** 10.3390/pathogens12050704

**Published:** 2023-05-12

**Authors:** Sindy Burgold-Voigt, Stefan Monecke, Anne Busch, Herbert Bocklisch, Sascha D. Braun, Celia Diezel, Helmut Hotzel, Elisabeth M. Liebler-Tenorio, Elke Müller, Martin Reinicke, Annett Reissig, Antje Ruppelt-Lorz, Ralf Ehricht

**Affiliations:** 1Leibniz-Institute of Photonic Technology (Leibniz-IPHT), 07745 Jena, Germany; 2InfectoGnostics Research Campus, 07743 Jena, Germany; 3Institute for Medical Microbiology and Virology, Dresden University Hospital, 01307 Dresden, Germany; 4Department of Anaesthesiology and Intensive Care Medicine, University Hospital, 07747 Jena, Germany; 5Thuringian State Authority for Food-Safety and Consumer Protection (TLLV), 99947 Bad Langensalza, Germany; 6Friedrich-Loeffler-Institut (Federal Research Institute for Animal Health), Institute of Bacterial Infections and Zoonoses, 07751 Jena, Germany; 7Friedrich-Loeffler-Institut, Institute of Molecular Pathogenesis, 07743 Jena, Germany; 8Institute of Physical Chemistry, Friedrich Schiller University, 07743 Jena, Germany

**Keywords:** *Staphylococcus aureus*, enterotoxin E, bacteriophage, wildlife, microarray, NGS

## Abstract

*Staphylococcus* (*S.*) *aureus* colonizes up to 30% of all humans and can occasionally cause serious infections. It is not restricted to humans as it can also often be found in livestock and wildlife. Recent studies have shown that wildlife strains of *S. aureus* usually belong to other clonal complexes than human strains and that they might differ significantly with regard to the prevalence of genes encoding antimicrobial resistance properties and virulence factors. Here, we describe a strain of *S. aureus* isolated from a European badger (*Meles meles*). For molecular characterisation, DNA microarray-based technology was combined with various next-generation sequencing (NGS) methods. Bacteriophages from this isolate were induced with Mitomycin C and characterized in detail by transmission electron microscopy (TEM) and NGS. The *S. aureus* isolate belonged to ST425 and had a novel *spa* repeat sequence (t20845). It did not carry any resistance genes. The uncommon enterotoxin gene *see* was detected in one of its three temperate bacteriophages. It was possible to demonstrate the induction of all three prophages, although only one of them was expected to be capable of excision based on its carriage of the excisionase gene *xis*. All three bacteriophages belonged to the family *Siphoviridae*. Minor differences in size and shape of their heads were noted in TEM images. The results highlight the ability of *S. aureus* to colonize or infect different host species successfully, which can be attributed to a variety of virulence factors on mobile genetic elements, such as bacteriophages. As shown in the strain described herein, temperate bacteriophages not only contribute to the fitness of their staphylococcal host by transferring virulence factors, but also increase mobility among themselves by sharing genes for excision and mobilization with other prophages.

## 1. Introduction

*Staphylococcus aureus* can cause a variety of pyogenic infections, sepsis and toxin-mediated conditions, such as food intoxications, staphylococcal scalded skin syndrome (SSSS) or toxic shock syndrome (TSS) [1,2]. Although it is a common hospital- or community-acquired pathogen of high medical relevance, it is not restricted to human hosts. Cases of infection or colonisation with *S. aureus* have been described from a wide range of wild [3,4,5,6,7,8,9,10,11] and domestic animals [12,13,14,15,16,17]. Recent studies indicated that wildlife strains of *S. aureus* frequently belong to other clonal complexes (CC) than human strains and that there is a partial overlap only to strains from domestic animals. Many wildlife lineages, even from Western Europe, are poorly known yet [6]. In general, the odds of transmission of such strains to humans appears to be low. The prevalence of genes encoding antimicrobial resistance properties is currently low and most wildlife strains of *S. aureus* are not likely to serve as reservoir for resistance genes [6,18,19,20]. One exception is the beta-lactam resistance gene *mecC*, which is frequently found in *S. aureus* from small mammals (CC130, CC425 and CC599), especially of hedgehogs (*Erinaceus europaeus*) [6,21,22,23] in which penicillin-producing dermatophytes maintain a selective pressure favouring beta-lactam resistance [24,25,26].

The wide range of different hosts observed for *S. aureus* in general and for many of its lineages in particular might indicate a high versatility of *S. aureus*. Indeed, it harbours a wide spectrum of similar, apparently redundant virulence determinants. For instance, there are more than 20 different superantigens/enterotoxins, seven distinct bicomponent leukocidins, five to six exfoliative toxins, and six haemolysins [27,28,29,30]. This diversity could be a reason for the ability of *S. aureus* to successfully colonise or infect different host species.

Many of the relevant virulence factors of *S. aureus,* such as some of leukocidins mentioned above, or some enterotoxins, are localised on prophages [27,31,32,33]. Prophages are bacteriophage genomes integrated into the bacterial host genome as a latent form of a releasable phage. By release and reinfection of other staphylococcal hosts, they can be spread horizontally across *S. aureus* populations and by transmitting genes encoding virulence factors, they can drive bacterial evolution [34,35,36].

An infection of a *S. aureus* clone by a bacteriophage might determine and influence its host specificity. For instance, *S. aureus* strains that carry prophages with genes *sea* (enterotoxin A), *chp* (chemotaxis-inhibiting protein), *scn* (staphylococcal complement inhibitor), and/or *sak* (staphylokinase) in various combinations tend to cause disease in humans, while isolates from the same lineages but without prophage can be found in animals [33]. Likewise, the four distinct phage-borne leukocidins, *lukF/S-PVL*, *lukF-P83/lukM*, *lukP/Q* and *lukF/S*-BV are associated with infections in specific hosts, i.e., humans, cattle and small ruminants, horses [37] or beavers [29], respectively. Genes located on bacteriophages are more mobile than others and therefore, it would be interesting to analyse the carriage of virulence factors with regard to the host species of the staphylococcal strains in question.

Given the poor knowledge on animal lineages of *S. aureus*, even from Western Europe, data for such an analysis are still not sufficient. A combination of DNA microarray technology as quick and economic screening and molecular typing tool combined with subsequent next-generation sequencing (NGS) for detailed characterisation is a promising toolbox for that task. One aim of the study was to prove and confirm the feasibility of this approach.

Badgers have previously been known to occasionally carry *S. aureus*. An earlier study on European wildlife identified four isolates among 29 examined animals [6]. Three were identified as CC425 (a lineage also known from various animals, see below and [6,38,39,40,41,42,43,44,45] and one was assigned to CC25 (previously found in humans, [46,47,48]). A recent study from Spain [49] did not identify *S. aureus* in the twelve badgers examined, and another one from Northern Ireland failed to detect MRSA in badgers [50]. In analogy to their role as reservoir of bovine tuberculosis [51,52], badgers could transmit wildlife strains of *S. aureus* to domestic animals via contamination of pasture, and humans could eventually be affected via contamination of milk or meat products either by infection or by ingestion of enterotoxins. However, to increase the knowledge on the role of *S. aureus* in wildlife and its importance for transmission to humans, here we characterized a *S. aureus* isolate from a European badger.

First, a microarray-based approach was applied which allowed to screen for all relevant virulence factors, adhesins and antibiotic resistance genes of *S. aureus*, and MLST assignment was performed. Second, the isolate was subjected to Illumina and Oxford nanopore next-generation genome sequencing (NGS) identifying three prophages. All of them were inducible by Mitomycin C treatment [31,53]. The phage preparation was examined by transmission electron microscopy and the isolated phage DNA was analysed by NGS.

## 2. Materials and Methods

### 2.1. Case Report

A 9.5 kg adult male Eurasian badger (*Meles meles*) was found as roadkill in Nordhausen district in the German Federal State of Thuringia. Post-mortem examination revealed various circular ulcers on the skin of the neck, head, and legs attributable to superinfected squamous cell carcinoma. Additionally, an inflammation of kidneys and a hyperplastic tumour of the spleen were found. *S. aureus, Clostridium perfringens* type A and Lancefield Group C streptococci were cultured from kidneys by standard microbiological techniques. Microbiological tests for *Salmonella* species (sp.) were negative. Parasitological findings included the detection of *Protostrongylus* sp. in the lungs, as well as of *Capillaria* sp. and coccidiae in the bowel contents. Fluorescent antibody tests for rabies and distemper were negative.

### 2.2. Amplification, Labelling, and Microarray Analyses

The *S. aureus* isolate V40 was characterised using the DNA microarray based Interarray *S. aureus* kit (fzmb GmbH, Bad Langensalza, Germany). Primer and probe sequences are listed in Appendix A. Protocols and procedures were in accordance with the manufacturer’s instructions. Purified DNA was used in a linear primer elongation using one primer per target. Thus, targets were amplified simultaneously while incorporating biotin-16-dUTP into the resulting single-stranded amplicons. Amplicons were stringently hybridised to the microarray, washed and incubated with a horseradish-peroxidase-streptavidin conjugate. After further incubation and washing, hybridisations were detected by adding a locally precipitating dye. An image of the microarray was taken and analysed using the InterVision reader, software and database (fzmb GmbH, Bad Langensalza, Germany).

### 2.3. Whole Genome Sequencing by Illumina and Oxford Nanopore Technology (ONT)

Whole-genome sequencing of *S. aureus* strain V40 was carried out using both, short-read Illumina and long-read ONT technology, in order to obtain genome completeness. For Illumina sequencing, an Illumina HiSeq2500 genome analyser (Illumina, Essex, UK) was used. Sequencing reads (approx. 300 bp) were assembled de novo using SPAdes version 3.10.1. In order to avoid fragmentation of the genome sequence across different contigs and to unambiguously characterise its prophages, the isolate was additionally sequenced using Oxford nanopore technology (ONT), i.e., MinION flow cells (FLO-MIN106D, containing R9.4.1 pores). Prior to library preparation, the genomic DNA was purified with the AMPure bead kit (Agencourt AMPure XP, Beckman Coulter, Krefeld, Germany) and potential nicks were repaired by using a combination of NEBNext FFPE DNA Repair Mix and NEBNext Ultra II End repair/dA-tailing Module (New England Biolabs, Ipswich, MA, USA). The incubation times were tripled. Pre-processed genomic DNA was used for library preparation with the ligation kit SQK-LSK109 (ONT) and the native barcoding expansion kit EXP-NBD114 (ONT), followed by a second purification step using AMPure beads. The ligation of sequencing adapters was again followed by a third purification step with AMPure beads. The library was completed by adding sequencing buffer and loading beads. An initial flow cell quality check showed 1196 active pores at the start of the sequencing. Flow cell was loaded with 80 ng (measured by Qubit 4 Fluorometer; ThermoFisher Scientific, Waltham, MA, USA) of genomic DNA from the prepared library. The sequencing run for 72 h using the MinKNOW software version 21.06.0.

### 2.4. MLST and Spa Typing

MLST based on sequences of housekeeping genes *arcC, aroE, glpF, gmk, pta, tpi* and *yqiL* as well as *spa* typing were initially performed according to previously published protocols [54,55] using publicly available databases (https://pubmlst.org/bigsdb?db=pubmlst_saureus_seqdef&page=sequenceQuery, http://spa.ridom.de/, accessed on 26 April 2023). Results were checked and confirmed using data from next-generation sequencing.

### 2.5. Phage Induction and Phage DNA Preparation

Phage induction was performed as described previously [29,56,57]. In short, overnight cultures of bacteria were inoculated in 2xTY medium and cultivated at 37 °C until the middle of the exponential growth phase (t = 2 h, OD = 0.60). Mitomycin C (Roche, Basel, Switzerland) was added to a final concentration of 0.5 μg/mL and cultivation was continued at 30 °C until the optical density (OD at 600 nm), compared to the previous measurement point, started to decrease. This usually occurred after ca. 6 h of cultivation. The lysate was centrifuged at 4 °C and 3000× *g*, and the supernatant was neutralized with 0.1 N NaOH and filtered with a 0.20 µm cellulose acetate (CA) membrane filter (Sartorius, Göttingen, Germany). In order to isolate phage DNA (p-DNA), the phage filtrate was again centrifuged for 30 min at 4 °C and 3000× *g*. The resulting supernatant was first treated with 10 µg/mL DNAse I (Sigma Aldrich, Steinheim, Germany) and 10 µg/mL RNAse (QIAGEN, Hilden, Germany) 1 h at 37 °C. Then, 20 mM EDTA, 50 µg/mL proteinase K and 0.2% SDS were added sequentially and incubated for another hour at 65 °C and 300 rpm. Phenol-chloroform extraction was subsequently performed as described previously [58]. Phase lock gel light tubes (Quantabio, Beverly, NJ, USA) were used in each step for better separation of the phases. Finally, isolated DNA was concentrated in a SpeedVac vacuum concentrator (Eppendorf, Hamburg, Germany) at 1400 rpm, at room temperature (20 °C) for 25 min. The final concentration was measured using the Qubit 4 Fluorometer (ThermoFisher Scientific, Waltham, MA, USA) according to the manufacturer’s instructions.

### 2.6. Sequencing of Phage DNA Applying ONT

ONT sequencing of the isolated p-DNA was performed on a MinION Flongle flow cell (FLO-FLG001 with an R9.4.1 pore). The library was prepared using the 1D genomic DNA by ligation kit (SQK-LSK109, ONT for flongle) as per the manufacturer’s instructions with slight modifications. Initially, a bead (Agencourt AMPure XP, Beckman Coulter) clean-up step was conducted before library preparation. The g-TUBE shearing step was skipped, and instead, NEBNext FFPE DNA Repair Mix and NEBNext Ultra II End repair/dA-tailing Module (New England Biolabs, Ipswich, MA, USA) were used to repair any potential nicks in DNA and DNA ends in a combined step. To ensure thorough repair, the incubation time was doubled. Following this, a second AMPure bead purification step was performed, and sequencing adapters were ligated onto prepared ends. Finally, the library was purified a third time using AMPure beads, and ONT sequencing buffer and loading beads were added. Prior to sequencing, an initial quality check of the flongle flow cell (ID: AES761) indicated 84 active pores. The library was loaded with a concentration of 31.6 ng/µL (measured by Qubit 4 Fluorometer; ThermoFisher Scientific, Waltham, MA, USA), totalling approximately 930 ng. The sequencing ran for 24 h and the MinKNOW software version 20.06.5 was used.

### 2.7. Bioinformatic Analysis of Sequencing Data

For both, bacterial and p-DNA, the guppy basecaller (v4.4.1 for phage and 4.4.2 for bacterial DNA sequencing runs, Oxford Nanopore Technologies, Oxford, United Kingdom) translated and trimmed the MinION raw data (fast5) into quality tagged sequence reads (4000 reads per fastq-file). To obtain a smaller, better subset of reads, Filtlong (v0.2.0 for phage, 0.2.1 for bacterial sample) was used with a median read quality of 15 and a minimum read length of 10,000 bp. Flye (v2.8.2 for phage and v2.8.3 for bacterial sample) was used to assemble this subset of reads to high quality contigs (parameter: min-overlap 1000 bp, --meta and --plasmids). Then, a racon-medaka pipeline (four times racon v1.4.19 for phage and v1.4.2 for bacterial sample; once medaka v1.2.0 for both) was applied for polishing. For medaka, the models r941_min_high_g360 (phage) and r941_min_sup_g507 (bacterial sample) were used. The NCBI Prokaryotic Genome Annotation Pipeline (PGAP version 2021-01-11. build5132) was used to annotate all assembled contigs, in addition to a manually curated in-house database of *S. aureus* genes.

### 2.8. Phage Detection by Transmission Electron Microscopy (TEM)

Negative staining was carried out on the phage preparations as described in detail [29]. In brief, copper grids filmed with formvar, coated with carbon and hydrophilized by glow discharge were placed on drops of phage preparations for 30 min. After washing with distilled water, one grid of each preparation was contrasted with 1% phosphotungstic acid and one with 1% uranyl acetate for 1 min. Grids were examined in a transmission electron microscope (Tecnai 12, FEI Deutschland GmbH, Dreieich, Germany) and representative micrographs were taken with a digital camera (TEMCAM FX416, TVIPS, Gauting, Germany). Particle size was measured using the EM-Measure software (TVIPS).

## 3. Results

### 3.1. Typing and Strain Characteristics

The *S. aureus* isolate V40 belonged to ST425 (18-33-6-20-7-50-48), *agr* group II, capsule type 5, and had a novel *spa* repeat sequence (Ridomt20845: 14-44-12-17-23-44-12-17-17-17-17-23-24).

The array hybridisation profile is shown in Appendix A, genes identified by sequencing in Appendix A. In short, the isolate did not carry any known resistance genes. The SCC*mec* XI element and the *mecC* gene were absent. Virulence factors included the staphylococcal enterotoxin E gene, and an unnamed ubiquitous enterotoxin homologue (CP000046.1, locus tag SACOL1657; FR821779.1, locus tag SARLGA251_15050), as well as leukocidin genes *lukD/E.* Genes for staphylokinase (*sak*), staphylococcal complement inhibitor (*scn*), enterotoxin A (*sea*), and chemotaxis inhibitory protein (*chp*), which in human isolates are usually associated with haemolysin beta integrating phages, were absent. The isolate contained biofilm-related genes *icaA/C/D* and a set of microbial surface components recognising adhesive matrix molecules (MSCRAMMs), including *bbp, clfA, clfB, ebh, ebpS, eno, fib, fnbA, fnbB, map, sdrC, sdrD* and *vwb*, but *cna* (encoding collagen adhesin) and *sasG* (Staphylococcal protein G) were not detectable.

### 3.2. Comparison to Other CC425 Sequences and Strains

We identified 23 sequences of ST425 strains in publicly available databases (Appendix A). In general, all ST425 are highly uniform matching our isolate in all core genomic features, although a variable presence of *fnbB, sdrD* and *vwb* was noted. The superantigen-like gene *ssl09* was not detected in isolate V40. One published sequence (SAMEA1708877) apparently lacked both, *hla* and the *setB* locus.

With regard to toxins, thirteen ST425-MSSA sequences and all six ST425-MRSA-XI sequences did not carry any enterotoxins. Three MSSA carried *tst1* and enterotoxin genes *sec* and *sel* (SAMEA1708766, SAMEA2298547, SAMEA2298548). Only one MSSA sequence (SAMN03219992; JXIG01000629.1) was found to harbour *see* (locus tag QU38_13900). This strain originated from a food sample from Switzerland [59]. The authors characterized in an earlier study [6] 29 ST425 isolates from European wildlife (badger, red fox, wild boar, red deer and roe deer) out of which three isolates were positive for *see*. Among badgers, however, two out of three were positive (including the one characterized here in detail).

Phage-borne leukocidin genes *lukF*-P83/*lukM* have been observed in some ST425-MSSA isolates from Italian Red deer [45,60] but neither in published sequences nor in the badger isolate.

While the present isolate was negative for SCC*mec*-associated genes, six published ST425 sequences (SAMN04537353, SAMN04537353, SAMEA1904132, SAMEA1708930, SAMEA2272771, SAMEA1033305 and SAMEA3491693) carry SCC*mec* XI elements and indeed, ST425 was, together with CC130, one of the first lineages described to carry that element including *mecC* [39,61]. One genome sequence (SAMEA1708769) also included an unknown SCC element harbouring *ccrA/B-1* but lacking *mecA/C* genes. Two others contained cadmium resistance genes that might be associated with SCC elements (see below).

The beta-lactamase operon *blaZ/R/I* was found in three of the published genome sequences ((SAMEA1708766, SAMEA2298547, SAMEA2298548). Among the 29 wildlife isolates mentioned [6], only one was positive for *blaZ*. In addition, those strains with SCC*mec* XI elements carry a distinct SCC*mec*-XI-associated beta-lactamase gene (FR821779.1, locus tag SARLGA251_00250). A tetracycline resistance gene, *tet*(K), was present in two of the published sequences (SAMEA2298547, SAMEA2298548). Heavy metal resistance genes in ST425 sequences, including the study isolate, include chromosomal genes *arsB* (arsenical pump membrane protein) and *arsC* (arsenate reductase), as well as *czrB* (zinc and cobalt transporter protein). In addition, ST425-MRSA-XI harbour an arsenic resistance operon associated with SCC*mec* XI, and two different sets of cadmium resistance genes (one, possibly SCC-associated, in SAMEA1708883, SAMEA2298571, another in SAMEA1708769, see Appendix A) were identified. The studied isolate did not carry these genes.

### 3.3. Phage Integration Sites and Sequences

The V40 genome sequence (Appendix A) contained three different bacteriophages, and all three of them also were recovered from and sequenced in phage suspensions, resulting from treatment with Mitomycin C (Figure 1, Appendix A). All three phage genomes (Appendix A) could be assigned to the *Siphoviridae* based on homologies to previously published sequences as well as on morphological features of the TEM images, and they showed the typical modular structure of *Siphoviridae* [62].

One prophage (phiSa-V40-1) was identified that spanned 43,246 nucleotides. Most notably, it contained the excisionase gene, *xis,* which the other phages lacked. It integrated between the rho-independent terminator of *rpmF/isdB* and *isdB* (corresponding to GenBank FR821779.1; SARLGA251_10410 in LGA251 and CP097571.1; M8789_05270 in this strain). This is also an integration site known for other phages (for instance, in *S. aureus* Newman, GenBank AP009351.1, locus tags NWMN_0991 to NWMN_1039).

A second prophage (phiSa-V40-2) is integrated into the beta-haemolysin gene *hlb*, which is another well-known integration site for prophages [33,63,64]. However, the genes usually associated with *hlb*-converting phages in isolates from humans (*sea*, *scn*, *chp*, *sak*) were neither detected by array hybridisation nor by genome sequencing. This prophage had a length of 42,282 nucleotides.

Finally, a third prophage (phiSa-V40-3) contained the enterotoxin E gene, *see*. Its genome had a length of 46,121 nucleotides. It integrated into a gene encoding a putative protein, A5IT17 (CP097571.1; locus tags M8789_07145/M8789_07480), which is known to contain the attachment site of PVL-phages in strains such as USA300 (CP000255.1; locus tags SAUSA300_1380/SAUSA300_1440) or USA400-MW2 (BA000033.2; locus tags MW1377/MW1443). In the CC425 reference genome of LGA251, there is no phage in that region (GenBank FR821779.1; between SARLGA251_13970 and 13990), and the gene encoding A5IT17 is also absent. That strain lacks *see* and phage-borne leukocidin genes. In the other CC425 sequence that harboured *see* (SAMN03219992; JXIG01000629.1, [59]), a fragment of the *see* phage is also associated with a truncated A5IT17 (locus tag QU38_13875).

The CC425 reference genome of LGA251 harboured yet another prophage localised between *sufB* (SARLGA251_07750) and Q2YWM5 (SARLGA251_08370), but in the study strain V40, this region was not occupied by a prophage.

Sequencing of the phage preparations prepared by Mitomycin C treatment yielded very different coverages; 623 for the *xis*-positive phage (phiSa-V40-1), 54 for the beta-haemolysin-converting phage (phiSa-V40-2) and only 12 for the *see*-phage (phiSa-V40-3).

### 3.4. Phage Morphology

In the suspension resulting from Mitomycin C treatment, a high number of phages was observed by transmission electron microscopy (Figure 2). All phages had the morphology of *Siphoviridae* of the order *Caudovirales,* i.e., they had icosahedral heads and long, non-contractile, thin tails which were straight or slightly curved (Table 1, [62,65,66]). Tails had a stacked disc appearance as a result of spiral twisting. Distinct base plates were seen at the end of tails in many virions. Measurements of length and diameter of heads in a total of 79 phages from two preparations allowed to discern three types of phages (Table 1):
Type I phages (Figure 3A) with head length and head diameter being of approximately equal length (difference ≤ 5 nm)Type II phages (Figure 3B) with head length > head diameter (difference ≥ 6 nm)Type III phages (Figure 3C) with head length < head diameter (difference ≥ 6 nm)

The three types of phages were found with different frequency at a ratio of 12 Type I:3 Type II:2 Type III phages.

## 4. Discussion

To our present knowledge, this is the first detailed characterisation of a *S. aureus* isolate from a European badger (*Meles meles*). The isolate belonged to ST425/CC425. This is a clonal complex that has previously been described as a widely disseminated, highly virulent rabbit strain [38], but there were no further genotyping data available so that especially the toxin gene carriage of these rabbit isolates cannot be compared to our badger isolate. This lineage has also been found in red deer, roe deer, wild boar, red fox and cinereous vulture [6,40,41,42,43,45]. In addition, *mecC*-MRSA from this lineage has been identified in cattle [39].

The issue of host specificity of distinct lineages of *S. aureus* (and other infectious agents) is not only of academic interest but of high relevance. Currently, domestic animals vastly outnumber wildlife species in terms of both head-count as well as biomass [67]. While pathogens from domestic animals might endanger vulnerable wildlife populations [68,69,70], wildlife lineages, or mobile genetic elements, such as bacteriophages, from wildlife lineages might spill over into domestic animals and consequently into human populations. Emerging *mecC*-positive *S. aureus* strains (CC130, CC425, CC599) could be an example as well as *Mycobacterium bovis*, with the latter one being a commonly quoted example of the danger that badgers might pose to public and animal health [51,52]. More data on *S. aureus* lineages and on other potentially zoonotic pathogens in European wildlife species are needed to assess the current epidemiological situation as well as to monitor possible ecological changes.

The most conspicuous finding in the study isolate was the presence of the enterotoxin E gene *see* on an inducible prophage. Enterotoxin E is a superantigen that has been implicated in food poisoning outbreaks in humans [71,72,73], and that indeed acts emetic in monkeys [71]. Compared to other enterotoxin genes in *S. aureus*, this gene is very rare. Out of a collection of ca. 30,000 isolates from humans and various animals that all have been genotyped by microarray ([6,74] and author’s unpublished data), it was only identified in twelve isolates. Two of them originated from badger, one from roe deer [6], two from unspecified domestic ruminants and from human (*n* = 6) or unreported (*n* = 1) hosts. Isolates belonged to CC425-MSSA (*n* = 5), CC395-MSSA (*n* = 3; including NARSA-111; https://www.beiresources.org/Catalog/bacteria/NR-45917.aspx, accessed on 26 April 2023), CC395/ST1093-MSSA (*n* = 2; including one isolate from a blood culture of a patient with suspected Lyell syndrome) or CC88-MSSA (*n* = 1). Only one (out of 18) published CC425 genome sequence (see above) was also found to carry a *see* that also was phage-borne, with the phage being integrated at the same site as in the studied isolate. Regarding un-typed strains, *see* was commonly found in mastitis cattle from Italy [75] as well as in goats from Chongqing, China [76], possibly indicating the presence of endemic strains or local outbreak situations. In general, there are not enough data available on geographic distribution and presence in specific hosts; therefore, no conclusions on epidemiology and host specificity of that toxin can currently be drawn.

The genome of the *S. aureus* strain described herein contained three prophages, although two of these three prophages, among them the *see*-carrying phage, did not carry a known *xis* gene. As previously described [77], it is not uncommon for prophages to lose their ability to excise and to become incomplete or defective prophages that remain permanently embedded in their host genome [66]. However, we were able to demonstrate by sequencing the isolated phage DNA and by transmission electron microscopy that all three prophages could be induced. It might be possible that one of the many “hypothetical/putative proteins” encoded by the prophages or a gene product from an un-annotated conserved region neighbouring integrase genes in *Siphoviridae* (which is also present in, e.g., the PVL phage of USA300-FPR3757; CP000255.1; [1,590,604..1,590,734]) might act as excisionase.

Another assumption was that the gene product of the functional *xis* gene of prophage phiSa-V40-1 resulted in the ability of the other two prophages to excise from the host as well. The comparison of the coverage of the three prophage sequences with the results of TEM suggests that the bacteriophages with the icosahedral morphology are the *xis* gene-carrying bacteriophages, just because of their relative abundance compared to the phages with other phenotypic appearances. However, it should be noted that the TEM separation of bacteriophages into three groups might be artificial, being possibly affected by degenerate bacteriophages or preparation artefacts. Sequencing of the isolated phage DNA proved that all three prophages were induced, indeed, and that the phage with the *xis* gene was induced at a higher level than the other phages, resulting in a 10-fold higher sequencing coverage. Thus, it can be assumed that the presence of a *xis* gene allows phage induction with high efficiency and that phages without *xis* still can be excised, albeit at a lower level, provided the presence of the *xis* gene product in the host cell that might be encoded by another prophage. This has been observed not only for prophages; it is even possible that functional, temperate phages induce chromosomal *S. aureus* pathogenicity islands (SaPIs), facilitating an exchange of virulence and resistance factors via horizontal gene transfer [62,65,66]. This suggests that in the presence of one excisable phage, other prophages do not need to maintain their *xis* genes. Bacteriophages have been described as “selfish replicators” [78] as they naturally depend on parasitizing their hosts’ replication and metabolic machinery. Apparently, they also parasitize each other utilizing the excisionase encoded by other prophages.

The fact that they introduce additional virulence factors into the genome of their host bacterium and that one bacterium’s phages use shared proteins could indicate a transition from parasitism to some kind of commensalism resulting in a reduction in the fitness costs they impose on their bacterial host [79]. Seen from a practical point of view, it should be considered if the medical use of substances that elicit a stress response and induce phages [80,81,82] might thus contribute to an accelerated evolution of pathogens by facilitating the exchange of genetic information via SaPIs or bacteriophages.

In conclusion, this study describes *S. aureus* from yet another host species, the European badger. The isolate belonged to a lineage, CC425, that is widespread among wild and domestic animals of different species but that, apparently, does not play a role in infecting or colonizing humans. The ability to successfully colonize or infect different host species is multi-factorial. Most conspicuously, the isolate carried a rare enterotoxin gene (*see*). This factor is known to be associated with food poisoning in humans, but it is not known whether it plays a role in the pathogenesis in non-human hosts of *S. aureus*. As with many other virulence factors, it can also be exchanged via mobile genetic elements, such as, in the given case, bacteriophages.

## Figures and Tables

**Figure 1 pathogens-12-00704-f001:**
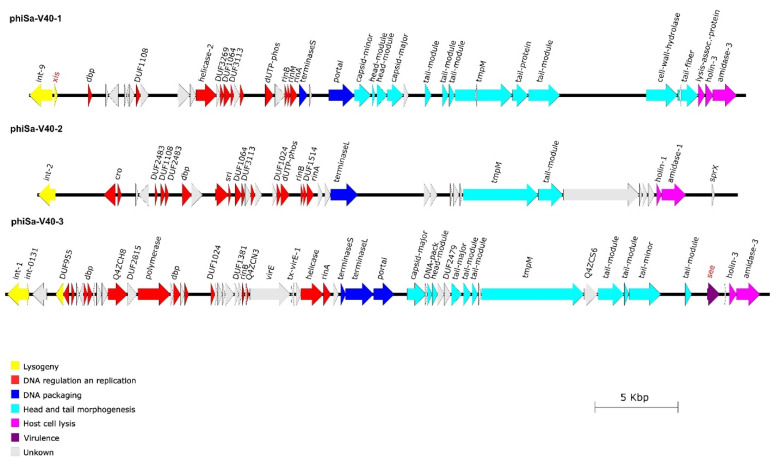
Graphic representation of the genome structures of *Siphoviridae* from V40. The direction, gene length, and position of aligned genes are shown by arrows. The colours represent the typical functional modules of *S. aureus Siphoviridae*. Phage phiSa-V40-1 furthermore contained the excisionase gene *xis* which the other phages lacked. Phage phiSa-V40-3 harboured the virulence gene *see*, which encodes for enterotoxin E.

**Figure 2 pathogens-12-00704-f002:**
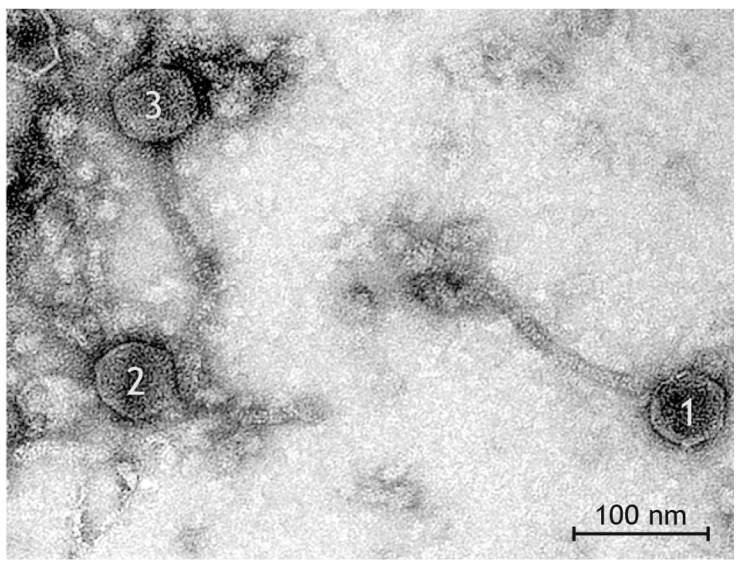
Transmission electron photograph of negative contrast preparation with uranyl acetate. High numbers of phage particles can be detected after Mitomycin C treatment of the badger isolate V40. The three types of bacteriophages and additional tail fragments are present in this image section: a Type I phage with isometric head (1), a Type II phage with mildly elongated head (2), a Type III phage with broad head (3).

**Figure 3 pathogens-12-00704-f003:**
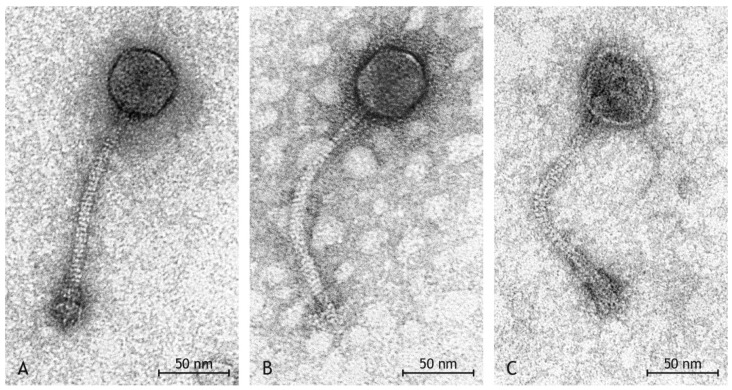
(**A–C**). Transmission electron photograph of negative contrast preparation with uranyl acetate. Higher magnification of the three types of phages. (**A**), Type I phage with icosahedral, length~diameter head, thin, non-contractile, slightly curved tail with stacked disc appearance and distinct base plate. (**B**), Type II phage with icosahedral, length > diameter head and thin, non-contractile, slightly curved tail with stacked disc appearance. The base plate is missing. (**C**), Type III phage with icosahedral to round, length < diameter head, thin, non-contractile, slightly curved tail with stacked disc appearance and distinct base plate.

**Table 1 pathogens-12-00704-t001:** Summary of morphological findings of transmission electron microscopy.

Group	Head Shape	Head Size	Tail Length	Tail Diameter	Base Plate Size	Number in Preparation
I	icosahedral	Ø 57 nm × 56 nm	Ø 176 nm	Ø 10 nm	Ø 31 nm × 24 nm	24
II	elongated	Ø 62 nm × 52 nm	Ø 195 nm	Ø 10 nm	Ø 32 nm × 24 nm	6
III	broad	Ø 57 nm × 66 nm	Ø 157 nm	Ø 9 nm	Ø 30 nm × 20 nm	6

## Data Availability

The genome sequence of the study isolate was submitted to GenBank, accession number CP097571.1. All other data are provided within the manuscript or its accompanying Appendix A.

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
