# Peer review of "Characterisation of a Staphylococcus aureus Isolate Carrying Phage-Borne Enterotoxin E from a European Badger (Meles meles)"

_pathogens, 2023, doi:10.3390/pathogens12050704_

Round 1

Reviewer 1 Report

The authors have worked on characterizing a strain of S. aureus isolated from badger. On identification of the strain they compared it with 23 sequences of the same strain available in literature to look at its core genomic features and any discrepancies. One of the primary findings was the identification of the enterotoxin E gene see on one of the three prophage. The authors also identified the absence of functional xis gene from two of the prophage.

There are a few opportunities to improve the clarity of the manuscript.

Page 3 Line 103-109 – Could the prophage from the other bacteria i.e. Clostridium and streptococci affect the genetic composition of S. aureus species? Since the badger was a road kill, it is possible that the bacterial infections occurred after the death and badger was not a living host. Also it would be helpful if the authors could talk in the discussion if the presence of multiple bacterial strains is common in wild animals

Page 3 Line 114 – It would be helpful if the authors could add the primer and probe sequences to this manuscript. It could go in the supplementary data.

Page 3 Line 118 – Please mention the wash buffer used and the dilution of hrp-streptavidin conjugate used for incubation.

Page 3 Line 138 – Please mention the sequence of adapters.

Page 9 Line 359 – Has enterotoxin from see gene shown to have an effect on animals or it is toxic only to humans?

Reviewer 2 Report

The manuscript submitted by Burgold-Voigt et al. presents results of genomic characterization of an S. aureus isolate in which three prophages were detected, one of which carries the see gene, encoding an enterotoxin of broad relevance and limited described. Overall, the article is well written and presents the results clearly. An extensive comparison with available genomes was performed, further highlighting the importance of see gene detection. After minor adjustments, I believe that the manuscript will be able to be accepted for publication. Some recommendations are described in the attached PDF file.

Reviewer 3 Report

Line 19: The use of abbreviation S. for Staphylococcus may be avoided, as it is used in the standard form throughout the literature. Additionally, not all the Staphylococcus colonizing humans are causing infections; please restructure the sentence.

Discussion: For further improvement of the manuscript, authors may write few sentences about broader implications of manuscript findings like clinical relevance of the identified virulence factors, epidemiological significance of the isolate's genetic profile etc.

 The study is well designed and include molecular, microbiological and bioinformatics techniques and findings. The only limitation is use of single isolate of S. aureus, which may not be true representative of entire population. The study's findings may only be applicable to the specific isolate analyzed and may not be applicable to other S. aureus strains.
